# Application of Fuzzy and Rough Logic to Posture Recognition in Fall Detection System

**DOI:** 10.3390/s22041602

**Published:** 2022-02-18

**Authors:** Barbara Pȩkala, Teresa Mroczek, Dorota Gil, Michal Kepski

**Affiliations:** 1Institute of Computer Science, University of Rzeszów, 35-310 Rzeszów, Poland; mkepski@ur.edu.pl; 2Department of Artificial Intelligence, University of Information Technology and Management, 35-225 Rzeszów, Poland; tmroczek@wsiz.edu.pl (T.M.); dgil@wsiz.edu.pl (D.G.)

**Keywords:** precedence indicator, knowledge measure, fuzzy inference, rule induction, posture detection, aggregation function

## Abstract

Considering that the population is aging rapidly, the demand for technology for aging-at-home, which can provide reliable, unobtrusive monitoring of human activity, is expected to expand. This research focuses on improving the solution of the posture detection problem, which is a part of fall detection system. Fall detection, using depth maps obtained by the Microsoft Kinect sensor, is a two-stage method. We concentrate on the first stage of the system, that is, pose recognition from a depth map. For lying pose detection, a new hybrid FRSystem is proposed. In the system, two rule sets are investigated, the first one created based on a domain knowledge and the second induced based on the rough set theory. Additionally, two inference aggregation approaches are considered with and without the knowledge measure. The results indicate that the new axiomatic definition of knowledge measures, which we propose has a positive impact on the effectiveness of inference and the rule induction method reducing the number of rules in a set maintains it.

## 1. Introduction and Description Problem

Fuzzy [1] and rough [2] sets provide tools for the analysis of significant imperfections of data and knowledge. The former allows classification of objects as belonging to a given degree to a set or relation. The latter provides approximations in cases where the information is incomplete. In this paper, we demonstrate how the mentioned theories can be merged into a hybrid system to improve the solution of the posture detection problem, which is a part of a fall detection system.

Considering that the population is aging rapidly, the demand for assistive technology for aging at home which can provide reliable, unobtrusive monitoring of human activity is expected to expand. One important aim of assistive technology is to provide prolonged independent living in a safe, home like environment without changing everyday lifestyle. Falls are a severe problem within the growing aging population. Many efforts have been undertaken to develop reliable methods of fall detection. The increasing number of studies in this area have allowed us to identify the major challenges and issues for fall detection technology, especially: performance, usability, and acceptance by the elderly. Fall detection systems need to be as accurate and reliable as possible both in terms of high sensitivity and specificity. In practice, this means that fall detectors must reliably distinguish between falls and activities of daily living (ADL) robustly, sustaining at low false alarm ratio. The method should not limit the placement of the sensors, or be sensitive to volatile environmental conditions. Such detection systems fall into two major categories, that is, wearable sensors and context-aware systems [3]. The main advantages of wearable sensors are size, usability, power consumption, and costs of use. The availability of cheap, embedded inertial sensors used in smartphones and smartwatches has contributed to the growth in their popularity in recent years. Usually, such approaches use threshold-based techniques to check if a person’s movement exceeds a predetermined threshold [4]. Some of the methods incorporate gyroscopes to obtain the person’s orientation [5]. Unfortunately, none of the above-mentioned methods provides satisfactory accuracy. Moreover, body-worn devices cannot be worn during certain activities, such as sleeping, changing clothes, and washing, moreover elderly people may forget to wear such devices. Context-aware systems are based on different kinds of sensors located in the user’s environment: cameras, microphones, pressure sensors, Doppler radar, and so forth. The main benefit of using context-aware systems is that no sensors need to be attached to the body of the monitored person, hence the reliability does not depend on the user’s willingness to wear the device. On the other hand, this form of activity monitoring is more expensive, invasive, and sometimes requires time to install and calibrate. Camera-based systems, which are one type of context-aware detectors, offer a promising way to detect falls and have been a subject of extensive research. Numerous attempts have been made to detect falls based on a single CCD camera, multiple cameras, stereo-pair cameras, and omnidirectional ones. Although CCD cameras offer several advantages, like the possibility to recognize various daily activities, the lack of ability to work in nightlight conditions and preserve privacy well may be considered serious drawbacks. Compared with the above-mentioned solutions, depth maps are insensitive to lightning conditions and provide 3D information that may substantially contribute towards the robust analysis of human activity.

This paper is focused on human pose recognition which is one part of the hierarchical system proposed in [6]. The mentioned system consists of two input fuzzy-reasoning engines (analyzing pose and movement separately) and a triggering alert Sugeno engine. The fuzzy reasoning on disjoint subsets of the linguistic variables performed by the engines leads to the reduction of the number of fuzzy rules needed for input-output mapping. Analyses of fuzzy and rough inference algorithms for posture detection, which are a part of the fall detection system, require methods that take into account uncertainty, for example, fuzzy set theory and rough set theory. These two theories model different types of uncertainty. The rough set theory takes into consideration the indiscernibility between objects. The second, that is, fuzzy set theory deals with the ill-definition of the boundary of a class through a continuous generalization of set characteristic functions. Given that these approaches pursue different goals, it is more natural to combine the two models of uncertainty than to force them to compete on the same problems. Thus, both approaches will be used in the proposed decision-making system.

The main objective of our research is to improve the solution to the posture detection problem. Therefore, a new hybrid system, based on fuzzy and rough sets, has been developed; the concept of the fuzzy information measure has been investigated and a new axiomatic definition of the knowledge measure has been introduced. In the system, two rule sets are investigated, the first one created based on a domain knowledge and the second induced based on the rough set theory, and two inference aggregation approaches are considered with and without knowledge measure. These measures together with various aggregation methods are used to evaluate the accuracy of the classification of rule sets in the decision-making process (the aim is also to indicate individual operators and fuzzification methods included in the tested system that meet the adopted assumptions, that is, to take into account the uncertainty represented by approximated values). The efficiency of the system is compared to [6]. The knowledge measure can be considered as a dual measure of fuzzy entropy or uncertainty. An entropy measure cannot capture all uncertainties in FSs. Knowledge measure has been studied in fuzzy environments, for example, in [7,8] and in intuitionistic fuzzy environments [9,10], which introduced knowledge measures in an IFS theory as a dual axiom system of intuitionistic fuzzy entropy. In this paper, the new knowledge measure is used to solve the problems of fuzzy inference (in a posture detection system) and tested using different aggregations in the process of aggregating premises. Its effectiveness is then compared using other measures known from the literature.

The following points summarize the main contribution of this study:(i)New measures:A new subsethood measure for fuzzy values is proposed and its validity is proved with the help of the example of use;A new knowledge measure for FSs is introduced and its significance is proved with the help of the example of use;(ii)A new hybrid system is proposed and used in a real decision making problem, i.e., a fall detection system for the elderly, in particular in a posture detection system:The proposed knowledge measure is applied to fuzzy inference problems;A rule induction method is applied to reduce the number of rules in a set while maintaining the effectiveness of the inference process and significantly improve the performance of a approximate reasoning.

The paper is organized as follows. In Section 2 related works are presented. In Section 3 methodology and data descriptions are proposed. In addition, elements of the fuzzy and rough sets theory as well as new measures of precedence and knowledge based on precedence indicators with their applications to fuzzy inference are presented. Finally, the experimental results of simulations of a hybrid approach to the fall detection problems are described in Section 4.

## 2. Related Work

Recently, depth cameras have been used in fall detection [11,12]. Ref. [13] applied the skeletal model obtained from Kinect SDK to fall detection. Ref. [14] proposed employing 3D joint tracking information to estimate the walking speed and to extract features describing the movements of a person going down the stairs. However, a person can be in one of many poses before a fall, so the skeleton extraction model may fail, or be unreliable during fall motion [15,16]. In [16] a two-stage fall detection method is proposed. Temporal segmentation of the vertical state time series of a person tracked in 3D is used in the first stage to identify on-ground events. In the second stage the confidence that the event was preceded by a fall is calculated, using a set of decision trees and features extracted from ground-based events. The improvement of fall detection reliability by combining depth and inertial sensors was proposed in [17]. Recent work demonstrates that merging the depth with accelerometer signal improves human activity recognition [18]. A more detailed overview of recent fall detection methodology using depth sensors is provided in [19]. Other approaches are based, for example, on convolutional neural networks (CNNs). However, due to the limited amount of data, their performance is limited. In [20] the authors used transfer learning where pre-traning on the ImageNet dataset AlexNet architecture was applied to accelerometric data, achieving an accuracy of 96.4%. Additionally, the authors of [21] also used depth data, however extracted from videos and thus applied to 3D-CNN. The detection of falls base on videos relies on multiple frames and uses more complex models, thus it can be considerably slower. By using data augmentation, they increased the model accuracy from 69.6% to 92.4% [22]. In this work we perform detection and classification of body contour on depth images. This approach ensures the privacy of the monitored person and is very effective in terms of processing speed. Our method involves merging the techniques mentioned above, fuzzy sets theory and rough sets theory. Despite the popularity of machine learning approaches, issues may arise with the use of simulated human fall event data. Firstly, the small number of actors, may not be sufficient to represent the entire population in terms of variability in human properties (i.e., height) or human biomechanics [23]. Scarcity of data may be problematic (especially for deep learning) so approaches other than traditional supervised classification are being investigated [24]. Another solution to address the lack of data is a customization of the parameters of the decision system to a person’s physical characteristics [25]. Our approach leverages the ease of customization and explainability of a fuzzy inference system by reducing the number of rules, allowing to build a linguistically understandable classifier maintaining high detection accuracy.

## 3. Methodology, Data, Theory and Tools Descriptions

For the purpose of to this article, we propose a new hybrid diagnostic system based on fuzzy and rough sets theory. To be specific, two rule sets are investigated, the first one created based on a domain knowledge and the second constructed by the rough set theory along with the main area of research which is concentrated on the concept of fuzzy information measure, and therefore the knowledge measure. These measures together with various aggregation methods are used to evaluate the accuracy of the classification of rule sets in the decision-making process.

### 3.1. Methodology and Data

The main goal of this research was to compare two approaches to posture recognition in fall detection: **I. Knowledge Approach** and **II. Rough Set Approach**. In the first approach a method based on a domain knowledge was used to generate a set of rules, the cardinality of which results from the combinatorial characteristic of this method. In turn, in the second approach induction method based on rough sets (described in Section 3.3) was used to reduce a set of rules. Next, both sets of rules were used in the fuzzy inference and evaluation process separately. Additionally, expert knowledge was used for modeling the selection of the parameters for the fuzzification function (described in Section 4). This combination of fuzzy and rough solutions is a novelty to the systems studied in the literature on fall detection problems. The concept of a hybrid approach (that we call a FuzzyRoughSystem, or FRSystem), presented in Figure 1, was based on three processes: *Data Acquisition Process*, *Fuzzy Inference Process* and *Evaluation Process*.

In the Data Acquisition Process, Kinect v1 cameras and an inertial motion sensor were used. The inertial sensors: PS Move and x-IMU collected data at 60 Hz and 256 Hz rates, respectively. The cameras were placed in different locations (one the front of the room parallel to the floor and the second one on the ceiling, facing down), in each case, the camera could be static or mounted on an active head. To preserve the user’s privacy, only the depth maps were analyzed. Depth maps were acquired using USB protocol, while accelerometric data were streamed wirelessly from the accelerometer using the Bluetooth protocol. For data acquisition, the OpenNI library was used, while the IMU sensor’s software was prepared based on the source codes provided by the manufacturer.

As a result, 5990 depth maps were collected in the UR Fall Detection Dataset. These depth maps were acquired using two Microsoft Kinect cameras from two different viewpoints. Each of the 30 distinct falls had about 150 labelled frames. The depth maps were stored as PNG16 images with 640 × 480 resolution.

The fall detection system, based on the images, was carried out in two stages: detection of a lying pose based on a single depth map and character movement analysis using dynamic transitions, however, in this work, we focused on the first stage of the system. Features describing the silhouette of a person at a given moment were determined as a result of the clustering of 600 images depicting characters in various poses, including during a fall and while performing ADL actions were analyzed. Ultimately, the following descriptors were selected from the set of features:H/W—the ratio of the height of the person’s bounding box to its width in the segmented point cloud.$H/H_{max}$—the ratio of the height of the person’s surrounding box in the current frame to the physical height of the person.$max(\sigma_{x},\sigma_{z})$—the maximum standard deviation of the values of points belonging to the character from its center of gravity along the axes of the Kinect camera coordinate system.$P_{40}$—the ratio of the number of points, lying no more than 40 cm above the floor, to the number of all points (belonging to the character point cloud).

Before we present and discuss the implementation of the new system (Section 4), we will recall some facts and introduce new elements in the fuzzy sets theory or rough sets theory.

### 3.2. Fuzzy Set Theory

Firstly, we recall the concept of a fuzzy set (relation) (cf. [26]). We consider fuzzy sets in a set P≠∅.

**Definition** **1**([1]). *An arbitrary operation R:p→[0,1] is a fuzzy set on P.*

All fuzzy sets on *P* will be denoted per FS(P) and the membership function describing the degree of belonging of p∈P to *R* is $\mu_{R}\left(P\right)$.

#### 3.2.1. Basic Operations

In this chapter, we will focus on the elementary operations (fuzzy negations and implication functions built on [0,1]) used in fuzzy reasoning, which is the basis of our novel system and which will also be recalled in Section 3.2.3.

**Definition** **2**(cf. [27]). *A non-increasing operation N:[0,1]→[0,1] which satisfies N(0)=1 and N(1)=0 is called a fuzzy negation N, which is strong if N(N(p))=p, p∈[0,1].*

**Example** **1**(cf. [28]). *Examples of fuzzy negations N are:*
*•$N_{k}\left(p\right)=1-p$ (strong negation called classical/standard negation);*

*•$N_{w}\left(p\right)={\left(1-p^{w}\right)}^{\frac{1}{w}}$, w>0;*

*•$N\left(p\right)=1-p^{2}$, which is strict but not strong;*

*•$N_{S}^{\lambda }\left(p\right)=\frac{1-p}{1+\lambda p}$, the Sugeno family of fuzzy (strong) negations, where λ∈(−1,∞) and for λ=0 we get the classical fuzzy negation.*


**Definition** **3**([29]). *An operation $I:{\left[0,1\right]}^{2}\to \left[0,1\right]$ which is a decreasing in the first component and increasing in the second component also fulfilling I(1,0)=0, I(0,1)=I(0,0)=I(1,1)=1 is called a fuzzy implication.*

Examples of fuzzy implications *I* are:Łukasiewicz implication—$I_{LK}\left(p,q\right)=\left\{\begin{array}{cc} 1, & \mathrm{if}\;\;p\leq q\\ 1-p+q, & \mathrm{otherwise}; \end{array}\right.$Fodor implication—$I_{FD}\left(p,q\right)=\left\{\begin{array}{cc} 1, & \mathrm{if}\;\;p\leq q\\ \max(1-p,q), & \mathrm{otherwise}; \end{array}\right.$Rescher implication—$I_{RS}\left(p,q\right)=\left\{\begin{array}{cc} 1, & \mathrm{if}\;\;p\leq q\\ 0, & \mathrm{otherwise}; \end{array}\right.$Reichenbach implication—$I_{RC}\left(p,q\right)=1-p+pq;$Kleene-Dienes implication—$I_{KD}\left(p,q\right)=\max\left(1-p,q\right).$

Now, we recall the basic and the most important operation on fuzzy sets, i.e., an aggregation function.

**Definition** **4**(cf. [30]). *An operation $A:{\left[0,1\right]}^{n}\to \left[0,1\right]$, n≥2 which is increasing and fulfils boundary conditions A(0,…,0)=0, A(1,…,1)=1 is called an aggregation function.*

**Example** **2.**
*Examples of aggregation functions are:*

*lattice: $T_{M}\left(p,q\right)=\min\left(p,q\right),\;\;S_{M}\left(p,q\right)=\max\left(p,q\right);$*

*algebraic: $T_{P}\left(p,q\right)=pq,\;\;S_{P}\left(p,q\right)=p+q-pq;$*

*Łukasiewicz: $T_{L}\left(p,q\right)=\max\left(0,p+q-1\right),$*



$S_{L}\left(p,q\right)=\min\left(1,p+q\right);$


*Arithmetic mean*

(1)
$$A_{mean}\left(p_{1},\ldots ,p_{n}\right)=\frac{1}{n}\left(p_{1}+\ldots +p_{n}\right);$$


*Geometric mean*

(2)
$$A_{gmean}\left(p_{1},\ldots ,p_{n}\right)=\sqrt[{n}]{p_{1}\ldots p_{n}};$$


*Square mean*

(3)
$$A_{2mean}\left(p_{1},\ldots ,p_{n}\right)=\sqrt{\frac{{p_{1}}^{2}+\ldots +{p_{n}}^{2}}{n}};$$


*The OWA operator (ordered weighted averaging) $OWA:{\left[0,1\right]}^{n}\to \left[0,1\right]$*

(4)
$$OWA\left(p_{1},\ldots ,p_{n}\right)=\sum_{i=1}^{n}w_{i}p_{\left(i\right)},$$


*(i) means a permutation of {1,…,n} such that $p_{\left(1\right)}\geq p_{\left(2\right)}\geq \ldots \geq p_{\left(n\right)}$ and $w=\left(w_{1},\ldots ,w_{n}\right)\in {\left[0,1\right]}^{n}$ is a vector of weights (i.e., $w_{i}\in \left[0,1\right]$ and $\sum_{i=1}^{n}w_{i}=1$) for $p_{1},\ldots p_{n}\in \left[0,1\right]$, n∈N.*


We will also employ the concept of pre-aggregation function [31], which satisfies the same boundary conditions as an aggregation function, but, in return to requiring monotonicity, directional monotonicity is needed, that is:

**Definition** **5.**
*An operation $F:{\left[0,1\right]}^{n}\to \left[0,1\right]$ is a pre-aggregation function if it fulfils*

*(1) There exists $\vec{r}\in {\left[0,1\right]}^{n}$$\left(\vec{r}\neq \vec{0}\right)$ a real vector which F is $\vec{r}$-increasing, that is, for all points $\left(p_{1},\ldots ,p_{n}\right)\in {\left[0,1\right]}^{n}$ and for all c>0 such that $\left(p_{1}+cr_{1},\ldots ,p_{n}+cr_{n}\right)\in {\left[0,1\right]}^{n}$, holds*

*$F\left(p_{1}+cr_{1},\ldots ,p_{n}+cr_{n}\right)\geq F\left(p_{1},\ldots ,p_{n}\right)$.*

*(2) F fulfils the boundary conditions: F(0,…,0)=0 and F(1,…,1)=1.*


**Example** **3**([31]). *Examples of pre-aggregation functions:*
*1. $F\left(p,q\right)=p-{\left(\max\left(0,p-q\right)\right)}^{2}$ is (0,1)—increasing (not an aggregation function).*

*2. $L_{\lambda }\left(p,q\right)=\frac{\lambda p^{2}+\left(1-\lambda \right)q^{2}}{\lambda p+(1-\lambda )q}$ (with convention 0/0=0) is (1−λ,λ)—increasing, for λ∈[0,1] (the weighted Lehmer mean).*


#### 3.2.2. Knowledge Measure

We will focus on an important measure, that is, the measure of fuzzification, that is, the knowledge measure. We propose to use this measure in the process of fuzzy inference when drawing conclusions from premises (in aggregating premises). Before we move on to a new idea of measuring knowledge in the fuzzy set environment/theory, we need to present a certain tool useful for the operation of fuzzy values, that is, a measure of inclusion of fuzzy values called a precedence indicator.

##### Precedence Indicator

Research on fuzzy sets began with the concept of Zadeh (1965), where K≤L iff $\forall_{p\in P}K\left(x\right)\leq L\left(x\right),$ but Bandler and Kohout (1980) proposed a new measure subsethood grade/precedence indicator of a fuzzy set in another fuzzy set which is based on a considering the infimum of an appropriate aggregation of implication operators. This idea of Bandler and Kohout inspired many authors to study fuzzy subsethood measures as the type of function σ:FS(P)×FS(P)→[0,1] with the different axiomatizations that have been proposed are not equal and they hinge on the examined applications. Based on this fact, and drawing inspiration from the works [32,33,34,35] in this paper we propose a new list of axiomatization for fuzzy precedence measure Prec:[0,1]×[0,1]→[0,1] as the class of implication operators which allows us to:Construct a new precedence indicator inspired by the axiomatic definition of the fuzzy subsethood measures;Construct new knowledge measures using a new precedence indicator;Apply new knowledge measures in fuzzy inference, as an illustrative example of the effectiveness of the proposed new measures.

**Definition** **6.**
*An operation $\mathrm{Prec}:{\left(\left[0,1\right]\right)}^{2}\to \left[0,1\right]$ is called a*
*
**precedence indicator**
*
*if it fulfils:*
**P1** 
*Prec(p,q)=0 iff p=1 and q=0;*
**P2** 
*Prec(p,q)=1 iff p≤q for any p,q∈[0,1];*
**P3** 
*If p≤q≤r, then Prec(r,p)≤Prec(q,p) and Prec(r,p)≤Prec(r,q) for any p,q,r∈[0,1].*



Now we propose the constructive method of the precedence indicator based on an aggregation and negation functions.

**Proposition** **1.**
*Let N denote a fuzzy negation (i.e., an antytonic operation that fulfils N(0)=1, N(1)=0) and A is the aggregation A≤max. Then*

(5)
$${\mathrm{Prec}}_{A}\left(p,q\right)=\left\{\begin{array}{cc} 1, & if\;\;p\leq q,\\ A\left(N\left(p\right),q\right), & otherwise \end{array}\right.$$


*is the precedence indicator.*


Here are some examples of the precedence indicators that satisfy Proposition 1.

**Example** **4.**
*For $A=A_{mean}$ and standard negation N we have*
*1* 

(6)
$${\mathrm{Prec}}_{A}\left(p,q\right)=\left\{\begin{array}{cc} 1, & if\;\;p\leq q,\\ \frac{1-p+q}{2}, & otherwise \end{array}\right.$$

*or for Sugeno negation with λ=1 we have*
*2* 

(7)
$${\mathrm{Prec}}_{A}\left(p,q\right)=\left\{\begin{array}{cc} 1, & if\;\;p\leq q,\\ \frac{1}{2}\frac{1-p}{1+p}+\frac{q}{2}, & otherwise \end{array}\right.$$

*for p,q∈[0,1].*



We pay attention to the fact that precedence indicators create a subclass of fuzzy implication functions as we observe in the following example.

**Example** **5.**
*The following operations are implication function but not precedence indicators:*

(8)
$$I\left(p,q\right)=\left\{\begin{array}{cc} 1, & if\;\;p\leq q,\\ 0, & if\;\;p=1,q\neq 1,\\ \frac{1}{2}, & otherwise, \end{array}\right.$$


(9)
$$I\left(p,q\right)=\left\{\begin{array}{cc} |p-q|, & if\;\;p<q,\\ 1-|p-q|, & if\;\;p=q,\\ A(N(p),q), & otherwise \end{array}\right.$$


*for p,q∈[0,1].*


##### Knowledge Measure

In this part of the work, we consider the crucial concept of information in the setting of uncertainty, that is, the idea of the knowledge measure of a fuzzy set, and suggest a new construction process for it by use of a precedence indicator. Cognitively, the knowledge measure is dual to the entropy measure of the arbitrary fuzzy set which gives the average values/height of fuzziness/ambiguity existing in the fuzzy set. Similarly, we can wonder about the average amount of knowledge present in the fuzzy set. Thus, a knowledge measure of a fuzzy set needs to satisfy the following axiomatic postulates. We propose some generalisation (in the fourth axiom) of the axiomatic definition of knowledge measure presented in [7,8].

**Definition** **7.**
*For R∈FS(P) a knowledge measure would satisfy the following properties:*
**K1** *K(R) has maximum value iff R is a crisp set, i.e., $R(p_{i})=0$ or* 1 *for all $p_{i}\in P$,***K2** 
*K(R) has minimum value iff R is the most fuzzy set, i.e., $R(p_{i})=0.5$ for all $p_{i}\in P$,*
**K3** 
*$K\left(R^{*}\right)\geq K\left(R\right)$, where $R^{*}$ is a crisped version (sharpened) of R,*
**K4** 
*$K\left(R\right)=K\left(R^{N}\right)$, where $R^{N}$ is the duality (complement) of set R for strong fuzzy negation N, i.e., $R^{N}\left(p\right)=N\left(R\left(p\right)\right),p\in P$ (for classic negation N we obtain a complement relation of R).*



We suggest the following construction method of the knowledge measure.

**Proposition** **2.**
*Let Prec be a precedence indicator that satisfies Proposition 1, where aggregation A is symmetric and N is the strong negation with an equilibrium point 0.5 (i.e., N(0.5)=0.5) for R∈FS(P), card(P)=n, n∈N, then*

(10)
$$K\left(R\right)=\frac{1}{n}\sum_{i=1}^{n}\frac{|\mathrm{Prec}\left(1,R\left(p_{i}\right)\right)-\mathrm{Prec}\left(R\left(p_{i}\right),0\right)|}{1-\min(\mathrm{Prec}\left(1,R\left(p_{i}\right)\right),\mathrm{Prec}\left(R\left(p_{i}\right),0\right))}$$

* is a knowledge measure.*


**Proof.** Let i=1,…,n. At the beginning let us note that 0≤K(R)≤1.(**K1**) is obvious with the assumption about *R*, Prec, and their properties. Because for a crisp relation of *R* we have:
for $R(p_{i})=1$Prec(1,1)=1 and Prec(1,0)=0 orfor $R(p_{i})=0$Prec(1,0)=0 and Prec(0,0)=1
and as consequence we obtain K(R)=1.Conversely, suppose K(R)=1, this is possible for$|\mathrm{Prec}\left(1,R\left(p_{i}\right)\right)-\mathrm{Prec}\left(R\left(p_{i}\right),0\right)|=1$ for all *i*, which implies$\left(\mathrm{Prec}\left(1,R\left(p_{i}\right)\right)=1\;and\;\mathrm{Prec}\left(R\left(p_{i}\right),0\right)=0\right)\;or\;\left(\mathrm{Prec}\left(1,R\left(p_{i}\right)\right)=0\;and\;\mathrm{Prec}\left(R\left(p_{i}\right),0\right)=1\right),$
so from P1 and P2 we obtain R(p)∈{0,1},p∈P, that is, *R* is crisp relation.(**K2**) By Proposition 1 and $R(p_{i})=0.5$ for all *i* and from the symmetry property of *A* and for the equilibrum point 0.5 of *N* we observe Prec(1,0.5)=A(N(1),0.5)=A(0,0.5)=A(0.5,0)=Prec(0.5,0), i.e., K(R)=0. Conversely, by assumption K(R)=0 we obtain$|\mathrm{Prec}\left(1,R\left(p_{i}\right)\right)-\mathrm{Prec}\left(R\left(p_{i}\right),0\right)|=0$ for all *i*, thus$\mathrm{Prec}\left(1,R\left(p_{i}\right)\right)=\mathrm{Prec}\left(R\left(p_{i}\right),0\right)$, which implies $R(p_{i})=0.5$ for all i.(**K3**) If $R^{*}$ is crisper than *R*, that is,$R^{*}\left(p_{i}\right)\geq R\left(p_{i}\right)$ for $R(p_{i})\geq 0.5$,$R^{*}\left(p_{i}\right)\leq R\left(p_{i}\right)$ for $R(p_{i})<0.5$.Based on Proposition 1 and for
$$\mathrm{Prec}\left(R^{*}\left(p_{i}\right),0\right)\leq \mathrm{Prec}\left(R\left(p_{i}\right),0\right),\;\mathrm{Prec}\left(1,R\left(p_{i}\right)\right)\leq \mathrm{Prec}\left(1,R^{*}\left(p_{i}\right)\right)$$
and
$$\mathrm{Prec}\left(1,R\left(p_{i}\right)\right)\geq \mathrm{Prec}\left(R\left(p_{i}\right),0\right)\;for\;R\left(p_{i}\right)\geq 0.5.$$Thus
$$|\mathrm{Prec}\left(1,R^{*}\left(p_{i}\right)\right)-\mathrm{Prec}\left(R^{*}\left(p_{i}\right),0\right)|\geq |\mathrm{Prec}\left(1,R\left(p_{i}\right)\right)-\mathrm{Prec}\left(R\left(p_{i}\right),0\right)|,$$
that is, $K\left(R^{*}\right)\geq K\left(R\right).$ In a similar way we consider the case $R(p_{i})<0.5$.(**K4**) Based on Proposition 1 we observe for the symmetric aggregation *A*:
$$|A\left(0,R^{N}\left(p_{i}\right)\right)-A\left(R\left(p_{i}\right),0\right)|=|A\left(0,R\left(p_{i}\right)\right)-A\left(R^{N}\left(p_{i}\right),0\right)|\;for\;all\;i,$$
as a consequence we have $K\left(R^{N}\right)=K\left(R\right)$, which completes the proof.    □

**Example** **6.**
*If in Proposition 2 we used precedence indicators satisfying Proposition 1 with $A\in \{A_{mean},A_{gmean},A_{2mean},\min,\max\}$ and N is standard (classical) negation, then we obtain knowledge measure K(R) for R∈FS(P).*


#### 3.2.3. Knowledge Measure and Fuzzy Inference (Mamdani)

The known and popular area of fuzzy logic and its extensions application is approximate reasoning, where from uncertainty/imprecise inputs/fuzzy premises or rules we often obtain uncertainty/imprecise inferences. Approximate reasoning has been used in many fields, for example, medical diagnosis, expert systems and control systems.

The main goal of this part of the paper is to explore the more general algorithm of approximate reasoning by using the general modus ponens property with the arbitrary aggregation functions next to the new knowledge measure. In the beginning, an algorithm for multi conditional approximate reasoning based on the new aggregation-based composition rules is proposed. The use of knowledge measure in fuzzy reasoning is a new accent in the classical model of inference. Thus we obtain a modification of the standard fuzzy reasoning method.

Approximate reasoning is the procedure where a possible uncertainty/imprecise conclusion is implied from a collection of uncertainty/imprecise premises. The classical modus ponens schema, was extended by Zadeh [36] to fuzzy reasoning in the following way and we obtained the GMP, that is, Generalized Modus Ponens:


Proposition:  If  p  is  D  then  q  is  E



Fact:      p  is  D’



----------------------------------------------------------



        q  is  E’,


where $E^{\prime }$ is the fuzzy set in the universe *Q*. The main plus of the GMP is that we can obtain new information even if $D^{\prime }$ and *D* are different. Usually, in the GMP the fuzzy rule is represented using a fuzzy relation *R* on the referential set P×Q. Existing different methods to build *R* can be used [37]. The most promising:

R(p,q)=I(D(p),E(q)), where *I* is an implication function. We may build the implication function from the aggregation function: I(p,q)=A(1−p,q) with A(1,0)=A(0,1)=1. Thus we can also create the relation *R* using the aggregation function by specific assumptions.

The fuzzy inference process is as follows
(11)$$E^{\prime }\left(q\right)=A_{p\in P}\;B\left(D^{\prime }\left(p\right),R\left(p,q\right)\right);\;i.e.\;E^{\prime }=D^{\prime }\circ R,$$
where A,B are aggregation functions on [0,1]. The basic inference process has the form presented in Figure 1.

Our novelty in the fuzzy inference in the process of aggregating premises is the proposal to use the combination of aggregation and knowledge measure as the following new operator:(12)$$O_{R}=B\left(A_{i=1}^{n}\left(p_{i}\right),K\left(R\right)\right),$$
where *R* is a fuzzy set on *P*, where cardP=n. Thus premises data in the given rule and *K* knowledge measure created by Proposition 2 and A,B are aggregation functions.

### 3.3. Rough Set Theory

The rough set theory use the indiscernibility relation to discover information about objects in an information system.

**Definition** **8**([38]). *An information system (IS) is an ordered quadruple (U,AT,V,f) where U is a finite nonempty set of objects, AT is a finite nonempty set of attributes, $V={⋃}_{a\in AT}V_{a}$; is a nonempty finite set of values of attributes, where $V_{a}$ is the domain of attribute a, and f:U×AT→V is an information function such that $f\left(x,a\right)\in V_{a}$ for all x∈U and a∈AT.*

A *decision table* is a type of information system. In the decision table the set AT=A∪D; A is a set of attributes, and D is set of decisions, D∩A=∅. Whereas, a *concept* is the set of all cases with the same decision value [39].

**Definition** **9**([2]). *For each subset of attributes A⊆AT a binary indiscernibility relation IND(A) on U can be determined as follows:*
IND(A)={(x,y)∈U×U|∀a∈A,f(x,a)=f(y,a)}.

Let a∈A, v∈V, and p = (a, v) be an attribute-value pair. The set [p] of all cases from U for which attribute a has value v is called a block of attribute-value pairs [40]. The rule induction Algorithm 1 LEM2 [39], in order to find a local covering of an input set, explores the space of attribute-value pairs.

Let X be a subset of U and P be a nonempty collection of nonempty sets of attribute-value pairs. The set P is a *minimal complex* of X if and only if X depends on P and no proper subset $P^{{}^{\prime }}$ of P exists such that X depends on $P^{{}^{\prime }}$ [39]. ρ is a *local covering* of X if and only if the following conditions are satisfied:each member P of ρ is a minimal complex of X,${⋃}_{p\in \rho }\left[P\right]=X$ρ is minimal [39].
**Algorithm 1** LEM2.**Input:** a set X**Output:** a single local covering ρ of set X X:=G; ρ:=∅; **while **
G≠∅
** do**     P:=∅     $P_{G}=\left\{p|\left[p\right]\cap G\neq \emptyset \right\}$     **while** P=∅ or [P]⊈X **do**         select a pair $p\in P_{G}$ such that |[p]∩G| is maximum;         if a tie occurs, select a pair $p\in P_{G}$ with the smallest cardinality of [p];         if another tie occurs, select first pair;         P:=P∪{p}         G:=[p]∩G         $P_{G}:=\left\{p|\left[p\right]\cap G\neq \emptyset \right\}-P$     **end while**     **for each** p∈P **do**         **if** [P−{p}]⊆X **then** P:=P−{p};         **end if**         ρ:=ρ∪{P};         $G:=X-{⋃}_{P\in \rho }\left[P\right]$;     **end for** **end while** **for each **
ρ∈P
**do**     **if** ${⋃}_{P^{\prime }\in \rho -\left\{P\right\}}\left[P^{\prime }\right]=X$ **then** ρ:=ρ−P;     **end if** **end for**

The LEM2 algorithm has been used successfully in many areas, recently in [41,42,43,44,45].

## 4. Implementation and Results

We implemented the inference system of FRSystem (Figure 1) in the following way: for the values of each input, that is, H/W, $H{/}_{max},$$max(\sigma_{x},\sigma_{z})$, $P_{40}$ we generated the fuzzy sets by using the adequate membership function needed for suitable rules, so for Lo (low value of the feature), Me (average value of the feature), Hi (high value of the feature) and the value of *isLy (lying position)*, *myLy (maybe lying position)* and *notLy (not lying position)* we use function type Z, Gaussian and type S, respectively (the Gaussian function is uniquely built by two different Gaussian functions). For the above functions we propose the following parameters:H/W:$\mu_{H/W}^{Lo}\left(p,0.5,1.25,2\right),\;\mu_{H/W}^{Me}\left(p,2,0.5,2,0,4\right),$$\mu_{H/W}^{Hi}\left(p,2,2.6,3.2\right);$H/max:$\mu_{H/max}^{Lo}\left(p,0.25,0.4,0.6\right),\;\mu_{H/max}^{Me}\left(p,0.6,0.1,0.6,0.2\right),$$\mu_{H/max}^{Hi}\left(p,0.6,0.8,1\right);$$max(\sigma_{x},\sigma_{z})$:$\mu_{max(\sigma_{x},\sigma_{z})}^{Lo}\left(p,260,285,310\right),$$\mu_{max(\sigma_{x},\sigma_{z})}^{Me}\left(p,310,17,310,33\right),$$\mu_{max(\sigma_{x},\sigma_{z})}^{Hi}\left(p,310,360,410\right);$$P_{40}$:$\mu_{P_{40})}^{Lo}\left(p,0.18,0.3,0.42\right),\;\mu_{P_{40}}^{Me}\left(p,0.42,0.08,0.42,0.09\right),$$\mu_{P_{40}}^{Hi}\left(p,0.42,0.55,0.68\right);$Pose:$\mu_{Pose}^{isLy}\left(p,0.22,0.36,0.5\right),\;\mu_{Pose}^{mayLy}\left(p,0.5,0.09,0.5,0.09\right),$$\mu_{Pose}^{notLy}\left(p,0.5,0.63,0.77\right).$

Based on the collected data, two rule sets were generated independently. The first one, a result of the Rough Set Approach, contained 44 rules: 10 rules for the pose notLy, 34 rules for the pose mayLy and 10 rules for the pose isLy. The second one, a result of the Knowledge Approach (FRSystem, Figure 1), contained 81 rules (${\left(3\;cases\left(functions\right)\right)}^{4features}$, [46]): 13 for the pose notLy, 52 rules for the pose mayLy and 16 rules for the pose isLy.

Next, in the *Fuzzy Inference Process*, a modified version of the basic Mamdani model was applied to obtain a posture decision (lying or not). Namely, in fuzzy inference, in the process of aggregating premises, a combination of aggregation and knowledge measure was used (new aspect by applying the operator $O_{R}$, see Section 3.2.3) constructed using a new precedence indicator. The effectiveness of the new measure was compared with the classic model without using the knowledge measure (the Section 3 and Section 4 in the FRSystem (Figure 2)) and also the effectiveness of applying different aggregations in the fuzzy inference process was analyzed.

To demonstrate the effectiveness of the proposed hybrid approach the following characteristics were used:accuracy
(13)$$\begin{array}{c} \mathrm{ACC}=\frac{\mathrm{TP}+\mathrm{TN}}{\mathrm{TP}+\mathrm{TN}+\mathrm{FP}+\mathrm{FN}}, \end{array}$$
where TP is the number of correct isLy classifications, TN is the number of correct notLy classifications, FP is the number of notLy classifications as isLy and FN the number of isLy classifications as notLyspecificity
(14)$$\begin{array}{c} \mathrm{SPE}=\frac{\mathrm{TN}}{\mathrm{TN}+\mathrm{FP}}, \end{array}$$precision
(15)$$\begin{array}{c} \mathrm{PRE}=\frac{\mathrm{TP}}{\mathrm{TP}+\mathrm{FP}}, \end{array}$$sensitivity
(16)$$\begin{array}{c} \mathrm{REC}=\frac{\mathrm{TP}}{\mathrm{TP}+\mathrm{FN}} \end{array}$$
in the *Evaluation Process*. Note that accuracy means how close a measurement is to the actual or expected value. The precision says how close the sets of measurements are to each other. The recall is characterized as the percentage of relevant results that are correctly classified by the used model, and specificity means the percentage of true negative results.

Finally, the rules used in inference (I. Rough Set Approach and II. Knowledge Approach) were assessed based on: the number of correct classifications of the rule, the effectiveness of the rule in the set and the effectiveness of the rule within the decision class.

We assumed that the effectiveness of the rule in the set can be expressed as follows:(17)$$\begin{array}{c} \frac{the\;number\;of\;correct\;classifications\;of\;the\;rule}{the\;number\;of\;objects\;in\;the\;set}. \end{array}$$

In turn the effectiveness of the rule within the decision class can be determined as follows:(18)$$\begin{array}{c} \frac{the\;number\;of\;correct\;classifications\;of\;the\;rule\;in\;the\;decision\;class}{the\;number\;of\;objects\;in\;the\;decision\;class}. \end{array}$$

Based on the above-mentioned measures, a rule ranking was created. First, the strongest rules from the set classification point of view were identified. Then, among the strongest rules, the ones which turned out to be the most effective within the decision class were selected. In this way, the rules that were critical to pose detection were indicated. The rules that were critical to pose detection were indicated. Finally, we use the center of gravity method for the defuzzification process.

To measure the effectiveness of our approach, the above-mentioned characteristics: accuracy (ACC), specificity (SPE), precision (PRE), and recall (REC) (sensitivity) were used. We studies the following cases:Determination of the effectiveness of classic fuzzy inference (without the knowledge measure and without the rule reduction) in fall detection problems, Table 1;Assessment of the impact of different aggregation functions and different knowledge measures, i.e., precedence indicators, on the effectiveness of classification of the reduced and nonreduced rules, using the FRSystem, Table 2;Verification of the effectiveness of the different knowledge measure construction methods in the FRSystem, proposed by us and others known from the literature, Table 3.Estimation of the effectiveness of each rule in the whole set and within the decision class.

Table 1, Table 2 and Table 3 show the experimental results obtained during the given dataset analysis. Presented outcomes in Table 1 maintain a high level of classification comparable to [6]. However, the next studies show that we observe progress in our classification results if we use the FRSystem (as can be seen in the result in Table 1 and Table 2) where the results are grouped for the original set of rules and after their selection by the rough method. Moreover, we compare the effectiveness of different aggregation functions used in the fuzzy inference, in the process of aggregating premises. We present the best results obtained for knowledge measures that satisfy Proposition 2 and are used in FRSystem. In particular, in $K_{1}$ we use in operator $O_{R}$ aggregation functions $A_{2mean}$ and B=F ($A_{2mean}$ used in the precedence indicator used in the Knowledge measure *K*) (which we denote as $K_{1}\left(A_{2mean},B_{F},A_{2mean}\right)$. Similarly, we created $K_{2}\left(A_{2mean},B_{F},\max\right)$, $K_{3}\left(\min,B_{F},A_{2mean}\right)$, $K_{4}\left(A_{mean},B_{F},A_{mean}\right)$, $K_{5}\left(A_{mean},B_{min},\max\right)$. In the presented results, we assume the results of each class we aggregate by the maximum.

The best results we obtained are marked in bold. As can be seen, the best performance is obtained for $K_{2}$ used in the FRSystem, with the following measures: ACC (96.9%), PRE (96.2%), SPE (87.8%) and REC (99.9%). What is more, we may say that the application of a reduced set of rules retained the classification level, that is, we obtained results with an acceptable difference of error, in a limit of the error at the level of about 0.01 (see Table 2). Thus, paths I and II in the FRSystem are comparable in the effectiveness aspect, but reducing the number of rules also has another important and positive effect on our model because we do not have to take into account all the attribute-value relationships. Only the most important relationships are selected in the induction process. A smaller and at the same time, optimal set of rules is easier for experts to evaluate.

Moreover, in Table 3 we compare our best results (we denote by *K* the knowledge measure built-in to the proposed method and used in the FRSystem) with other methods to build knowledge measures known in the literature (unlike our approach, the dependence (precedence indicator) of a given fuzzy value on the extreme (certain value) is not taken into account), such as: $K^{SLS}\left(F\right)=\frac{1}{n}\sum_{i=1}^{n}2\left[F^{2}\left(p_{i}\right)+{\left(1-F\left(p_{i}\right)\right)}^{2}\right]-1$ [8], $K^{AK}\left(F\right)=log_{2}\left[\frac{2}{n}\sum_{i=1}^{n}\left(F^{2}\left(p_{i}\right)+{\left(1-F\left(p_{i}\right)\right)}^{2}\right)\right]$ [7].

There, the fuzzy and dual values are taken into account while in our approach the given fuzzy value is compared by subsethood measure with the extremes (the largest and the smallest certain value), which gives a more complete picture of the uncertainty contained in the measurements. We observe the higher effectiveness of the proposed new knowledge measure (see Table 3). For comparison we take *K* from case $K_{2}$ from the result presented in Table 2:(19)$$\begin{array}{c} K\left(F\right)=\frac{1}{n}\sum_{i=1}^{n}\frac{|{\mathrm{Prec}}_{\max}\left(1,F\left(p_{i}\right)\right)-{\mathrm{Prec}}_{\max}\left(F\left(p_{i}\right),0\right)|}{1-\min({\mathrm{Prec}}_{\max}\left(1,F\left(p_{i}\right)\right),{\mathrm{Prec}}_{\max}\left(F\left(p_{i}\right),0\right))}, \end{array}$$
where for p,q∈[0,1] we have
$${\mathrm{Prec}}_{\max}\left(p,q\right)=\left\{\begin{array}{cc} 1, & \mathrm{if}\;\;p\leq q,\\ \max(N(p),q), & \mathrm{otherwise}. \end{array}\right.$$

In order to identify the most relevant attribute values (from a classification view point) for each decision class the rules were assessed first on the whole set, and then on the concepts. As a result, the values of the attributes clearly defining the detection of a lying or non-lying position are indicated and presented in Table 4. It should be noted that, the *H*/*W* attribute did not occur in the reduced set of rules, among the conditions of the most efficient rules for the notLy decision class. The absence of this attribute did not affect the quality of classification within this class in relation to the non-reduced set of rules. The remaining conjunctions of conditions for the most effective reduced and non-reduced rules were identical.

## 5. Conclusions

In this paper, we have provided the initial results of a very interesting new approach to the selection of appropriate aggregation functions and a set of rules for fuzzy inference in the problem of fall detection, especially posture detection. Moreover, the main research was concentrated on investigating the concept of a fuzzy information measure, presenting a new axiomatic definition for the knowledge measure, and using theirs in the proposed hybrid system. The results obtained for the mentioned aspects indicate the positive results of the new approach. Out of 81 rules (see [46]), by applying the LEM2 algorithm we indicate 44 rules (see [47]) which allow us to significantly reduce the dimensionality of the studied problem and facilitate its analysis while maintaining a high level of classification comparable to [6].

Our goal for future work is to develop this research on both theoretical and practical grounds. For example, we would like, in cooperation with an Elderly care home in Rzeszow, to expand the data set and develop some new methods to represent data, for example, a hybrid method that uses fuzzy and rough sets concerning uncertainty, so we will use interval-valued fuzzy set theory. In addition, the developed hybrid inference method seems to be very promising for use with different input data sets in the future. In particular, new measures of information may prove useful for the issues or methodologies observed in the works [7,8], where the proposed knowledge measure is utilized to calculate the weights vector, when weights are partially known and other when weights are completely unknown in economic terms, in multiple attribute decision-making methods or in image thresholding based on a fuzzy accuracy measure.

## Figures and Tables

**Figure 1 sensors-22-01602-f001:**
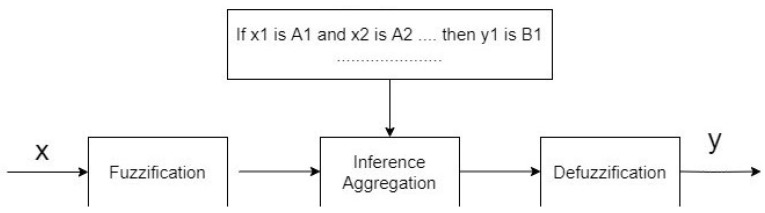
The FRSystem flowchart.

**Figure 2 sensors-22-01602-f002:**
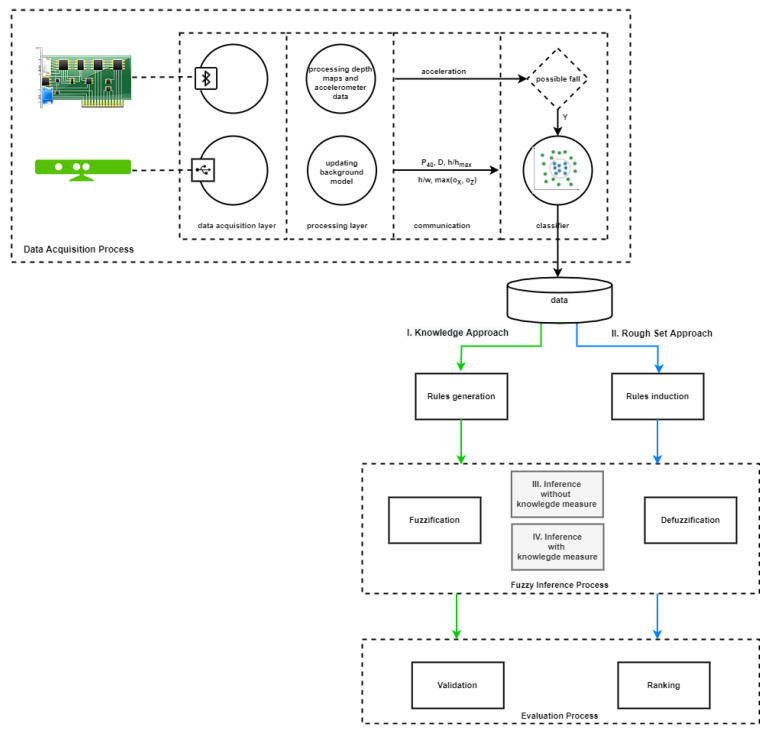
The Scheme of the fuzzy inference process.

**Table 1 sensors-22-01602-t001:** Confusion and classification evaluation metrics by the standard fuzzy inference system with aggregations from examples 2 and 3.

	$T_{m}$	$T_{p}$	$A_{\mathrm{mean}}$	OWA	F
TP	7303	7303	7420	7375	7420
TN	1968	1968	2069	2056	1969
FP	405	405	304	317	404
FN	149	149	32	77	32
ACC	0.944	0.944	**0.966**	0.960	0.956
PRE	0.947	0.947	**0.961**	0.959	0.948
REC	0.980	0.980	0.996	0.990	**0.996**
SPE	0.829	0.829	**0.872**	0.866	0.830

**Table 2 sensors-22-01602-t002:** Confusion and classification evaluation metrics with the operator *K* used in the FRSystem, where All and Red. means test on full and on reduced set of rules. respectively.

	$K_{1}$		$K_{2}$		$K_{3}$		$K_{4}$		$K_{5}$	
	**All**	**Red.**	**All**	**Red.**	**All**	**Red.**	**All**	**Red.**	**All**	**Red.**
TP	7443	7446	7442	7446	7430	7445	7440	7446	7440	7445
TN	2066	1938	2076	1956	2083	1939	2063	1961	2064	1985
FP	307	435	297	417	290	434	310	412	309	388
FN	9	6	10	6	22	7	12	6	12	7
ACC	0.968	0.956	**0.969**	0.957	0.968	0.956	0.967	0.957	0.967	**0.96**
PRE	0.960	0.945	**0.962**	0.947	0.962	0.945	0.96	**0.948**	0.96	0.95
REC	**0.999**	**0.999**	**0.999**	**0.999**	0.997	**0.999**	**0.999**	**0.999**	**0.999**	**0.999**
SPE	0.871	0.817	0.875	0.824	**0.878**	0.818	0.869	0.826	0.87	**0.84**

**Table 3 sensors-22-01602-t003:** Confusion and classification evaluation metrics with different knowledge measures used in the FRSystem.

	*K*	$K^{\mathrm{SLS}}$	$K^{\mathrm{AK}}$
TP	7442	7434	7444
TN	2076	2026	2064
FP	297	347	309
FN	10	18	8
ACC	**0.969**	0.963	0.968
PRE	**0.962**	0.956	0.960
REC	**0.999**	0.998	**0.999**
SPE	**0.875**	0.854	0.870

**Table 4 sensors-22-01602-t004:** Specification of the most relevant attribute values for decision classes, where Lo, Me and Hi means low, average and high value of the feature, respectively and Ly means lying position.

H/W	$H/H_{\max}$	$max(\sigma_{x},\sigma_{z})$	$P_{40}$	Concept
Hi	Hi	Lo	Lo	notLy
Me	Hi ∨ Me	Lo	Lo	
Lo	Lo	Lo ∨ Me	Hi	∼notLy

## Data Availability

The dataset used in this research work is on the website http://fenix.univ.rzeszow.pl/~mkepski/.

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
