# Peer review of "Application of Fuzzy and Rough Logic to Posture Recognition in Fall Detection System"

_sensors, 2022, doi:10.3390/s22041602_

Round 1
Reviewer 1 Report
This paper proposed a formulation or aggregation functions and rules inference in a posture recognition system called the FRSystem. It proceeds by comparing two approaches; Rough Set and Expert Knowledge. The formulations are then evaluated via a classification experiment for accuracy, specificity, precision, and recall.
Overall, the organization of the paper is very poor. The terms Rough Set and Fuzzy are used interchangeably, which should not be the case. Why does the expert knowledge approach still goes through the same process of fuzzy inferencing and ranking? Knowledge-based approach should be different methodology altogether. In addition the methods from the literature, KSLS and KAK, are also not deliberated.
It is difficult to assess the contribution when the methodology and evaluation is not clear. Finally, please observe some glaring typo errors:
Figure 2: Walidation
Line 352: Different font types of quotes
Line 421: TThe
Author Response
Dear Reviewer,
We are very grateful for all the stimulating comments, remarks, and suggestions which have helped to improve the quality of the paper.
Please see the attachment.
Best Regards, Barbara Pekala

Reviewer 2 Report
The authors presented their work related to posture monitoring through a hybrid method based on depth and accelerometer data. The results are encouraging and the research is relevant in the context of home monitoring systems for the elderly.
The paper needs to be properly checked for language mistakes, some were highlighted in the attached pdf.
Also, I highly suggest the authors to provide more information about how they collected the data, which Kinect version did they use, what IMU did they use, how were the cameras mounted in the room, and so on.

Author Response

(The authors gave the same response as above.)

Round 2
Reviewer 1 Report
The manuscript has been improved accordingly and is now structured to promote reader understanding. Upon reviewing the new version, please attend to the following comments.
Abstract, Introduction, Related Work: The main problem with these section are the lack of motivation and the absence of research gap. The authors stated the research focus, which is to improve the solution of the posture detection problem, without deliberating the problems and existing solutions. Related work of 1 paragraph is too short. This section only described existing work on the surface, without clear gap in the current techniques used for fall detection that calls for the need of FL and RS.
Methodology: Fig. 1 shows there is a ranking process during evaluation. Ranking is for evaluating the rules, not the fall detection. Furthermore, no ranking process/equation is described. Next, add a new Section 3.4 for knowledge measure and performance evaluation (ACC, SPE, PRE, and REC).
Results: Add discussion on how FL and RS achieved higher accuracy as compared to techniques in SAS and AK.
Other observations:
Line 267: Why the equations are both bulleted and numbered? Can you number all equations?
Line 301: Use proper indentation - it is unreadable
Line 372: Double quotes are of different font that suggests copy paste
Line 398: Section III and Section IV
Line 413: How the rule ranking works?
Line 429: Split the para. Presents different Ks as bullet list. Make a new equation for K similar to equations of KSLS and KAK in Line 452.
Overall, most notations in tables are not obvious i.e. OWA in Table 1, Red. in Table 2, Hi-Lo-Me in Table 4. All notations must be sufficiently described before or after the tables. Check for double use of parentheses and squared brackets in all citations. Check for proper capitalization as well.
Author Response
Dear Reviewer, Please see the attachment, best regards, Barbara Pękala

Round 3
Reviewer 1 Report
This paper is now suitable for publication.